Hybrid parking space prediction model: integrating ARIMA, Long short-term memory (LSTM), and backpropagation neural network (BPNN) for smart city development

Dahiya Anchal 1
Mittal Pooja 1
Sharma Yogesh Kumar 2
Lilhore Umesh Kumar 3 umeshlilhore@gmail.com
Simaiya Sarita 3
http://orcid.org/0000-0001-5913-5979 Haq Mohd Anul 4 m.anul@mu.edu.sa
http://orcid.org/0000-0002-7015-4131 Aleisa Mohammed A. 4
Alenizi Abdullah 5
1 Department of Computer Science & Applications, MDU , Rohtak, Haryana , India
2 Department of Computer Science and Engineering, Koneru Lakshmaiah Education Foundation, Green Field , Vaddeswaram, Guntur, AP , India
3 Department of Computer Science and Engineering, Galgotias University , Greater Noida, UP , India
4 College of Computer and Information Sciences, Majmaah University, Department of Computer Science , Al-Majmaah , Saudi Arabia
5 Department of Information Technology, College of Computer and Information Sciences, Majmaah University , Al-Majmaah , Saudi Arabia
Angiulli Giovanni
Electronic publication date: 2025 Jan 24
Publication date: 2025
Volume: 11
Electronic Location ID: e2645
Received 2024 Apr 18; Accepted 2024 Dec 13
Copyright: © 2025 Dahiya et al.
Copyright year: 2025
Copyright holder: Dahiya et al.
License: This is an open access article distributed under the terms of the Creative Commons Attribution License, which permits unrestricted use, distribution, reproduction and adaptation in any medium and for any purpose provided that it is properly attributed. For attribution, the original author(s), title, publication source (PeerJ Computer Science) and either DOI or URL of the article must be cited.
License URL: https://creativecommons.org/licenses/by/4.0/

Keywords: Smart city, Deep learning, ARIMA, Parking space, LSTM, IoT

Funding: Deanship of Scientific Research at Majmaah University R-2024-1419 This work was supported by the Deanship of Scientific Research at Majmaah University under Project No. R-2024-1419. The funders had no role in study design, data collection and analysis, decision to publish, or preparation of the manuscript.

==============================
Parking space prediction is a significant aspect of smart cities. It is essential for addressing traffic congestion challenges and low parking availability in urban areas. The present research mainly focuses on proposing a novel scalable hybrid model for accurately predicting parking space. The proposed model works in two phases: in first phase, auto-regressive integrated moving average (ARIMA) and long short-term memory (LSTM) models are integrated. Further, in second phase, backpropagation neural network (BPNN) is used to improve the accuracy of parking space prediction by reducing number of errors. The model utilizes the ARIMA model for handling linear values and the LSTM model for targeting non-linear values of the dataset. The Melbourne Internet of Things (IoT) based dataset, is used for implementing the proposed hybrid model. It consists of the data collected from the sensors that are employed in smart parking areas of the city. Before analysis, data was pre-processed to remove noise from the dataset and real time information collected from different sensors to predict the results accurately. The proposed hybrid model achieves the minimum mean squared error (MSE), mean absolute error (MAE), and root mean squared error (RMSE) values of 0.32, 0.48, and 0.56, respectively. Further, to verify the generalizability of the proposed hybrid model, it is also implemented on the Harvard IoT-based dataset. It achieves the minimum MSE, MAE, and RMSE values of 0.31, 0.47, and 0.56, respectively. Therefore, the proposed hybrid model outperforms both datasets by achieving minimum error, even when compared with the performance of other existing models. The proposed hybrid model can potentially improve parking space prediction, contributing to sustainable and economical smart cities and enhancing the quality of life for citizens.

Introduction

Smart city development is incredibly improving the living standards of the people. Despite gaining popularity, it faces several challenges due to rapid urbanization and limited resources. Thus, utilizing these limited resources most efficiently becomes highly important by proposing smart systems for all possible domains, including education, healthcare, business, parking, etc. Among all of these, smart parking systems have also become a need of the hour for providing solutions to a number of problems faced by users in smart cities (Singh, Jeong & Park, 2020). Such systems can directly target problems like traffic congestion, increased carbon emissions, fuel wastage, driver frustration, and many more.

The Internet of Things (IoT) is transforming cities into smart cities by increasing intelligence and connectivity. Various urban cities, including transportation systems, airports, hospitals, and shopping centers, are rapidly adopting IoT applications. With the number of IoT-connected devices projected to exceed 16 billion in 2023 (Simplio, 2023). The impact of IoT on information technology (I.T.) is truly remarkable. The advent of IoT technology has significantly simplified daily life, particularly in the past decade (Soe et al., 2020).

Parking management is an integral part of the development of any smart city where the IoT plays an important role by monitoring the available parking space and helping the users by providing real-time information about the space availability in the parking areas. Due to increase in vehicle ownership, parking has become one of the most challenging tasks in today’s society. As the number of vehicles increases, traffic congestion becomes a common problem faced by users. Searching for appropriate available parking spaces may lead to other problems like increased carbon emissions, fuel wastage, and driver frustration. Intelligent parking systems are becoming increasingly important due to the rising number of vehicles in urban areas. In recent years, parking challenges have gained significant attention due to the ever-increasing daily influx of vehicles onto the roads (Lin, Rivano & Le Mouël, 2017).

The existing parking management systems might produce acceptable outcomes temporarily, but the increasing number of vehicles on the roads means new solutions must be developed. With urban traffic challenges becoming more severe, innovative approaches are essential for efficiently handling parking areas and reducing the long-term pressure on drivers. These limitations motivated the proposal of a scalable model that can handle larger parking areas more efficiently (Dahiya et al., 2024).

Parking space prediction depends on the analysis of historical data as well as real-time updates for delivering accurate, reliable, and scalable parking solutions. These models are inefficient in handling long-term dependencies. Moreover, existing models cannot generalize across different environments, limiting their scalability, which may lead to inaccurate parking predictions. Thus, a more promising solution is required that can analyze different types of data: linear, nonlinear, and time series data for accurate predictions.

Thus, a novel hybrid model with two phases is proposed that will integrate the strengths of these existing techniques to solve the problems in existing models. In the first phase, data is preprocessed, and then auto-regressive integrated moving average (ARIMA) is used to analyze the linear trends efficiently from the time-series dataset (Amini et al., 2015). To handle nonlinear patterns, long short-term memory (LSTM) is also applied in the first phase (Gupta et al., 2022). In the second phase, backpropagation neural network (BPNN) is used to fine-tune the output layer and optimize overall model performance by reducing the error rate (Awan et al., 2020). The proposed hybrid model overcomes existing models’ limitations and provides accurate parking prediction with different time frames.

The main contribution is that the proposed hybrid model is scalable and can be deployed in smart cities, which will help in accurately predicting available parking spaces. The proposed hybrid model can better address short-term predictions with ARIMA’s help and long-term predictions with the help of LSTM. BPNN helps optimize the model and prevent overfitting, which will ensure robustness and adaptability. Other major factors are also considered during prediction, such as identifying significant variables that affect parking occupancy, the passage of time, and so on. All these factors are essential in developing a model to predict available parking spaces. The main contribution of this article is as follows:

(a) This article proposes a novel scalable hybrid model that combines ARIMA + LSTM + BPNNN to enhance prediction accuracy using an IoT-enabled time-series dataset.

(b) The proposed hybrid model enhances prediction accuracy by addressing non-linear patterns and data sparsity challenges.

The article is organized into six sections: In “Literature Survey”, a comprehensive literature survey of existing research on predicting parking spaces is conducted. In “Materials and Methods”, the algorithm of the proposed model is discussed. The implementation phase of the proposed hybrid model is detailed in “Implementation of the Proposed Hybrid Model”. “Experimental Results” presents the experimental results of the proposed model and compares the proposed hybrid model with existing models. In “Conclusion”, the article finally concludes and presents the future scope.

Literature survey

This section explores the existing work of researchers who contributed to the field of prediction of available parking spaces using sensor-generated data to make accurate predictions. Several researchers have contributed to this area to solve parking-related problems so that traffic congestion can be decreased and ultimately help in the development of sustainable environments for the citizens. This section will help establish a road map for researching this area (Yang et al., 2019).

This survey will help to find the gaps in the existing work. It will help to decide which models to combine so the prediction accuracy can be increased and that can handle the large datasets efficiently to manage parking space effectively. Table 1 presents the key findings of the literature survey.

Table 1 Key findings from the survey of the existing research work on parking space prediction.

Summary of the main insights and conclusions drawn from a review of recent research on parking space prediction models.

Author	Technique used	Findings	Future scope/Gap	
Amini et al. (2015)	ARIMA	Effective for linear trend predictions, but lacks adaptability	Non-linear data not captured efficiently that impact accuracy.	
Yang et al. (2019)	Graph convolutional neural network (GCNN) and LSTM	Various methods (linear regression, SVM, neural network, ARIMA) compared in different cities.	Investigate real-world implementation and fine-tuning of the proposed method.	
Kasera & Acharjee (2022)	LSTM	Good at modeling non-linear relationships, yet suffers from overfitting	Explore hybrid approaches to enhance prediction accuracy.	
Kaur, Roul & Batra (2023)	Hybrid deep learning (convolutional neural network (CNN) for feature extraction and extreme learning machine (ELM) for classification) for classification of parking spaces.	CNN-ELM is effective in real-time parking space detection, especially across varying weather conditions (sunny, overcast, rainy), tested on the PKLot dataset (approx.700,000 images).	While CNN-ELM performs well, the article does not explore its scalability across different cities or its adaptability to highly dynamic environments.	
Feng et al. (2023)	Deep gated graph recurrent neural network (G2RNN)	Captures spatial and temporal correlations in parking data, providing accurate predictions without gridding raw data.	Overfitting and scalability issues.	
Javadi, Moosaei & Ciuonzo (2019)	GPS, EKF, SVM, and wireless sensor networks (WSN)	Improved position accuracy using SVM, Public safety in smart cities.	Further enhancement of position prediction methods using machine learning algorithms, Continued research on WSN-based detection systems for enhancing public safety.	
Li, Li & Zhang (2018)	CNN and LSTM	Efficient cloud-based system using LSTM network for parking predictions.	Enhance real-time data integration and scalability for improved predictions.	

This extensive survey helps us to understand the gaps in the existing prediction models. It is clearly witnessed that existing models perform well in predicting smaller parking areas, but as the parking areas increase, the performance of predicted models decreases. There is a need for a novel approach that can handle both types of parking areas, smaller and larger. Moreover, more advanced approaches are needed to handle the complex data the sensors generate in the parking areas. These factors motivated us to propose a novel model that could address these issues and accurately predict parking space availability. The main aim of proposing the model is to make it scalable so that it will work on complex data efficiently by optimizing the results and making accurate predictions to improve urban traffic congestion. Various factors are considered, such as traffic patterns, historical data, etc. The scalable proposed hybrid model can efficiently predict parking space, reduce user search time, minimize road congestion, and improve urban planning.

Materials and Methods

This section describes different learning models used in the proposed hybrid model to enhance the performance of the prediction of parking space availability. “The proposed hybrid model consists of two phases. In the first phase, ARIMA and LSTM are applied, whereas BPNN is implemented to reduce the error rate in the second phase. The time series data generated by parking area sensors comprises linear and nonlinear values, making it complex and challenging to handle. In the proposed hybrid model, three different approaches are integrated to handle such complex data efficiently. This section discusses all three approaches, including the ARIMA statistical approach followed by the LSTM model (Chaturvedi et al., 2022). Then, BPNN is discussed to improve the accuracy of the model.

Auto regressive integrated moving average (ARIMA)

ARIMA handles the linear components of time series data effectively, modeling trends and seasonality in the short term. The ARIMA model predicts future values of the dataset based on its historical values and observed patterns (Medina-Salgado et al., 2022). ARIMA model uses a statistical method to predict available parking space in the time series dataset. It combines two components, autoregressive (AR) and moving average (MA), to predict future values based on past observations. Equation (1) is used to predict the Time-series dataset:

(1) y(t)=c+∑(φ(i)∗y(t−i))+∑(θ(i)∗ε(t−i))+ε(t).

In Eq. (1), y(t) represents the value of the time series dataset at time t, c is used as constant, φ(i) is used as lags coefficient for autoregressive value, θ(i) is used for moving average and ε(t) is used for error at time t.

Long short-term memory model (LSTM)

LSTM handles nonlinear patterns in the time series dataset and is the ideal choice for predicting available parking spaces. LSTM can retain information for extended periods (Awan et al., 2020). This is due to their proficiency in identifying and handling unknown temporal lags between critical data occurrences (Begum et al., 2022). The three primary gates that make up the fundamental architecture of an LSTM are the input (IN), output (OP), and forget gates. A memory cell plays a crucial role in the LSTM’s ability to store and preserve prior knowledge. It is a hybrid with the ARIMA model because it can handle complex, non-linear data patterns efficiently. It particularly addresses the limitations of the ARIMA model, known for its proficiency with linear data, providing a more sophisticated approach to predicting parking space occupancy (Gupta et al., 2022).

Back propagation neural network (BPNN)

BPNNs consist of an input layer, an output layer, and one or more hidden layers. It propagates errors from the output layer to hidden layers, allowing for weight adjustments and reducing prediction errors. It includes forward and backward propagation, similar to the backward error propagation method (Xiao et al., 2023). Error if occurs is sent back to the system in the opposite direction. This reverse process helps the model learn different aspects of the systems, such as weights and biases, to reduce errors and improve the prediction. This process is repeated until we get the desired result. This process helps the neural network to make accurate predictions.

Proposed hybrid model (ARIMA + LSTM + BPNN)

The proposed hybrid model is designed at two phases. In the first phase, data is preprocessed and fed into the proposed hybrid model that combines the ARIMA and LSTM models. ARIMA handles the linear components of the time series data, and LSTM handles the nonlinear components of the data. The LSTM model is hyper-tuned to get the best prediction from the model. Results obtained from the first phase are fed into the second phase, which uses BPNN to optimize the performance of the proposed hybrid model.

In training phase (Phase 1), linear trends in historical parking data are handled well with the ARIMA model. It allows the model to learn from past values. Further, nonlinear data which frequently appears in time series datasets is handled by LSTM, making it suitable for complex patterns in parking space usage. The predicted outcomes from Phase 1 are then passed to Phase 2 through the BPNN integrated at the output layer to optimize the predictions and fine-tuned the model’s output based on the learned patterns from ARIMA and LSTM. The key features of the proposed hybrid model are listed in Table 2.

Table 2 Key features of the proposed hybrid model.

The primary components and advantages of the proposed for parking space prediction.

Key feature	Characteristics	Observations	
ARIMA	ARIMA handles the linear components of time series data effectively, modeling trends and seasonality in the short term.	Suitable for predicting 1–24 h ahead, where linear trends are prominent.	
Differencing term is chosen as 1 (to remove seasonality).	
LSTM	LSTM captures complex nonlinear patterns and remembers long-term dependencies, ideal for time-series data like parking space availability.	LSTM model is best for predicting longer periods (24 h or more).	
Historical data fed: 50–100 time steps.	
Activation: Tanh and Sigmoid for capturing complex patterns.	
Optimize with BPNN	BPNN refines the accuracy by combining the outputs of ARIMA and LSTM. It adjusts the model based on errors to further improve accuracy.	BPNN with 3 hidden layers optimizes the combination of ARIMA and LSTM outputs.	
Learning rate: 0.9 for faster convergence.	
Historical observations used	ARIMA and LSTM require a sequence of past occupancy values to predict future availability.	50–100 time steps (historical observations) provide sufficient data to capture seasonality and nonlinear patterns.	
Future prediction window	The model predicts how many parking spaces will be available at future timestamps.	Predicts parking availability for 1–24 h ahead, ideal for both short-term and long-term forecasting.	
Attractive properties	ARIMA handles short-term linear trends well.	ARIMA captures daily and hourly trends.	
LSTM’s memory capability is ideal for capturing long-term dependencies.	LSTM models complex dependencies in the data.	
BPNN fine-tunes the combination of both models.	BPNN reduces residual errors from both models.	
Scalability of the model	The hybrid model can be adapted to various datasets and environments by adjusting parameters and the architecture.	Capable of processing larger datasets and adapting to different parking patterns by retraining with new data, maintaining performance across diverse urban settings.	

The above table highlights the key features of the proposed hybrid model. It highlights how combining all three approaches improves the overall accuracy and robustness of the parking space prediction. The pseudo-code representing the workflow of the proposed hybrid model is shown in the next section.

Pseudo code: proposed hybrid model for parking space prediction

The pseudo-code presented above is used to implement the proposed hybrid model. It provides a step-by-step description of different phases of the proposed hybrid model. The working model’s flow chart is presented in Fig. 1.

Figure 1 Proposed hybrid model (ARIMA + LSTM + Optimised by BPNN).

The architecture of the integrated framework for time series forecasting. The ARIMA module captures linear trends and seasonal components, providing a robust baseline.

The steps of the proposed hybrid model are outlined below:

Step 1: Preprocess the dataset. As we are working on the sensors dataset, it generates a lot of noise, so to get an accurate prediction; the first step is to preprocess the data to remove anomalies from the dataset.

Step 2: Parking space prediction dataset comprises a linear component (Lt) and nonlinear component (NLt). To analyze the dataset, the “statsmodels” package in Python is used to decompose the time-series dataset obtained from IoT sensors.

Decomposition aids in examining the seasonality, trend, and residual values of the feature employed for predicting parking space availability. Mathematical illustration of the raw dataset (yt) for parking space prediction is shown in Eq. (2).

(2) Yt=Lt+NLt.

Step 3: After observing that the dataset exhibits seasonality, the linear parts of the time-series dataset fed into the ARIMA model for predicting parking space availability. The predicted result of the ARIMA model, denoted as ( L¯t), is obtained using Eq. (3). The “pmdarima” package is employed to determine the optimal parameters for the ARIMA model in order to improve the accuracy of the parking space predictions.

(3) L^t=ARIMA(Yt)

Step 4: The residual quantities (NLt) of the ARIMA model are computed using Eq. (4) and then utilized as input for the LSTM model in the parking space prediction. Using ARIMA residuals as input for the LSTM model enhances prediction accuracy by allowing the LSTM to capture unresolved variations and complex patterns, creating a more robust prediction model.

(4) NLt=Yt−L^t.

Step 5: To process the nonlinear values of the time-series dataset, the LSTM model is employed, and Eq. (5) is used to predict more complex patterns in the dataset.

(5) NL^t=LSTM(NLt).

Step 6: The integration of ARIMA and LSTM forecasting values is optimized using BPNN. BPNN plays an important role in combining the linear and nonlinear predictive values represented by the function “f” in Eq. (6). BPNN optimizes the results to propagate error values backward in the neural network and adjust neuron weights in the forward direction, for enhancing the accuracy of parking predictions not achieved by existing work done by the researchers.

(6) Y^t=fL^t+NLt.

By utilizing BPNN, the model effectively predicts (Y^t) values by integrating the predicted results of ARIMA (Y^t) and LSTM (NLt). Unlike existing models, the proposed hybrid model (ARIMA + LSTM + BPNN) improved the prediction results obtained by the combination of ARIMA and LSTM. Instead, a BPNN is employed to establish the true functional relationship among the forecasted values of ARIMA and LSTM, highlighting the novelty of this existing model.

Implementation of the proposed hybrid model

This section will provide a detailed description of the implementation phase of the proposed hybrid model. This section first provides a detailed description of different datasets used to predict available parking spaces. Then, it provides the details of the Preprocessing steps that are used to implement the proposed hybrid model so that the dataset is ready for training and testing. Finally, the implementation phase will be presented, highlighting the steps used in training the model that will help implement the proposed hybrid model.

Dataset used

The Melbourne public dataset used to implement the proposed hybrid model is a comprehensive collection of data spanning two months, from January to February 2018 (City of Melbourne, 2024; https://data.melbourne.vic.gov.au/explore/dataset/on-street-car-parking-sensor-data-2018/information/). It encompasses various aspects of the city, including geographical information and parking sensor data. The dataset provides valuable insights into the parking situation in Melbourne during that time period (Zheng, Rajasegarar & Leckie, 2015).

First, the model is implemented on the Melbourne dataset. Further, to check the generalizability of the proposed hybrid model, it is again implemented on the Harvard dataset (https://dataverse.harvard.edu/dataset.xhtml?persistentId=doi:10.7910/DVN/YLWCSU), which includes both SFpark sensor data and Simulated Crowd-Sensing data. It collects data from different networks of sensors positioned in parking spaces around the city. The data includes information such as when a parking space is available, location, and occupancy status. Different attributes of the dataset include Block_Id, Block Start_Time, and many more. These two different models are used to check the proposed model’s performance in different environments to check the robustness of the proposed hybrid model.

Data preprocessing

The most important step before implementation is to preprocess the dataset so that the dataset is ready to make accurate predictions. Sensors generate a large amount of data that contains a lot of noise. As time-series dataset is used it need to be preprocessed, identifying the patterns from the dataset and then extracting the relevant features to enhance the accuracy of the proposed model (Kasera & Acharjee, 2022). Mean-filling is used to remove noise and missing values from the dataset. This method efficiently handles the missing values from the dataset. This step is essential for enhancing the accuracy of predicting available parking space, ultimately enhancing the predicting accuracy of the proposed hybrid model. The equation below is used to fill inconsistencies in the dataset.

(7) m¯(t)=1d∑i=1d⁡mi(t).

In Eq. (7), m¯(t) is used to find the average of the time period t, mi(t) is used as actual value and d as no. of days.

In certain parking slots, sensors generate the time-series data, as they are activated at 7:30 am and deactivated around 6:30 pm. This will make substantial fluctuations in the parking dataset. Therefore, the first and the last data points of each day are pruned to mitigate the problem of fluctuations in the parking dataset, which will ultimately result in more accurate analysis and enhance the predicting capability of the proposed hybrid model.

Time-series datasets contain different values, so the first task is to understand the seasonality, trends, and residuals from the available data. This step is necessary before the analysis and is necessary for accurately predicting available parking. A time-based splitting technique is used to recognize the temporal nature of the dataset. The proposed model’s performance is evaluated by employing systematic training and testing. The time-series dataset is divided into training sets and testing sets, and analysis is done in a different environment.

The parking dataset contains variations to ensure the proposed model’s performance (robustness and scalability) with data variation; both the datasets Melbourne and Harvard are divided into three different training and testing parts for implementation and evaluation purposes. In the first part, the dataset is divided into 70% training dataset and 30% testing dataset. In the second phase, the dataset is divided into different ratios, which are 80% for training the model and 20% for testing the model. This division provides a more extensive training set to the model so it can learn efficiently to predict the result. In the last phase, the dataset is divided into ratios, with 90% used for training and 10% used for testing the proposed hybrid model to predict available parking space.

In both of the datasets mentioned above, the proposed hybrid model consistently demonstrates the best performance with the lowest error values in different training and testing scenarios, indicating its superior predictive accuracy across different environments, as both datasets have different specifications. The main focus is maximizing the data used for training, which could result in a more refined and optimized model. By evaluating the proposed hybrid model across these different scenarios, we can comprehensively understand its performance and assess its effectiveness in predicting parking space availability. Optimizing its performance is essential to ensure the scalability of the proposed hybrid model.

Decomposing the parking dataset is necessary to identify better long-term trends, seasonal fluctuations, random noise, and many more. Firstly, both the parking datasets used to implement the proposed hybrid model must be decomposed. Decomposition helps us to determine the appropriate ARIMA model to use. Figure 2 illustrates the decomposed outcome of the Melbourne parking dataset used in this research. The decomposition analysis detects the presence of seasonality in the dataset, indicating recurring patterns over time but no discernible trend. Also, residual values are evident in the dataset, requiring careful consideration and handling in the modeling process.

Figure 2 Decomposition of Melbourne parking dataset.

The time series components of the dataset, including the observed values, trend, seasonal patterns, and residuals.

Figure 3 provides the same comprehensive view of the Harvard parking dataset as done for the Melbourne dataset. It will help to analyze parking occupancy fluctuates with different times, and it breaks complex data into more interpretable components. This decomposition can facilitate better prediction and help the planners to manage the parking resources. It will provide in-depth knowledge that will help identify the busy periods of different parking slots and guide the development of strategies to manage occupancy effectively, which will improve user experience and optimize resource allocation.

Figure 3 Decomposition of Harvard parking dataset.

The time series components derived from the dataset. The observed data represents the raw parking activity patterns.

We carefully examined both datasets using the Seasonal Decomposition of Time Series (STL) method. This method will break down the data into trends, seasonality, and residuals. With the STL decomposition, the trend part reveals the long-term shifts or variations in parking occupancy data. It allows us to spot any general trends over time. “When looking at parking occupancy, the seasonal part involves the patterns that happen at certain times, like daily, weekly, or monthly cycles. We use the ARIMA model to manage the sequential data in the seasonal dataset. We input both the testing and training data into this model to identify the fundamental patterns and trends. The results are shown in the next section after providing the training and testing data inputs to the ARIMA model.

Implementation phase

In this section, different phases of the proposed hybrid model implementation are discussed in detail. This section provides step-by-step training and testing results of the implementation of the proposed hybrid model for predicting available parking spaces.

ARIMA model

After decomposing both datasets, linear trends from the dataset are fed into the ARIMA model in Phase 1, as briefly discussed in “Data preprocessing”. To find the right parameters for the ARIMA model, we use the ‘auto arima’ function in the ‘pmdarima’ package. This function picks the best values for the parameters (p, d, q) and (P, Q, D, s) by looking for the lowest Akaike information criterion (AIC) value. Once executed on the training dataset, the “auto arima” function determines the selected parameters (p, d, q), (P, Q, D, s) as (2, 0, 0), (0, 1, [1], 4), with an associated AIC value of 520.056. These parameter values are crucial for accurately modelling and predicting the seasonal variations in the parking occupancy data using the ARIMA model.

Figures 4 and 5 display the ARIMA model’s diagnostic results on different datasets used for implementing the proposed hybrid model. The plot includes a standardized residual plot. The standardized residual plot indicates that the residual errors mostly revolve around zero, indicating the likelihood of error for the linear component of the data.

Figure 4 Diagnosis of ARIMA model to detect errors using the Melbourne dataset.

The diagnostic analysis performed to detect errors in the ARIMA model. This includes residual plots, ACF plots, and PACF plots, which evaluate the adequacy of the model.

Figure 5 Diagnosis of the ARIMA model to detect errors using the Harvard dataset.

The error analysis conducted to validate the ARIMA model.

Figures 6 and 7 represent the predicted results of the ARIMA model after training the model as discussed earlier, which are divided into three different scenarios.

Figure 6 Actual vs predicted results of the ARIMA model (Melbourne dataset).

The observed parking data with the predictions generated by the ARIMA model.

Figure 7 Actual vs predicted results of the ARIMA model (Harvard dataset).

The figure compares the observed parking data with the predictions made by the ARIMA model.

This section provides the predicted results of the ARIMA model. In the next section, LSTM is discussed in detail.

LSTM model

The residuals obtained from the ARIMA model are provided to the LSTM model in Phase 1 of the proposed hybrid model. LSTM model is hyper-tuned to predict the available parking space. It’s important to deal with the difference between the actual and predicted occupancy values to get precise results. The LSTM model bridges the gap and improves prediction accuracy. We split the residuals obtained from the ARIMA model into training, validation, and testing sets before adding them to the LSTM model. After that, we check the model’s performance and how it accurately predicts the available parking space. Additionally, we adjust the leftover numbers using the ‘MinMaxScaler()’ function to make training and prediction work better. MinMaxScaler() function will set the values between 0 and 1, helping the LSTM model understand the trends efficiently. The LSTM model uses adjusted leftover values as a starting point and figures out the hidden patterns to make better results. This method helps close the differences between the actual and estimated values, leading to more accurate parking space prediction.

Table 3 outlines the hyperparameters and settings for an LSTM model. The optimizer algorithm used during training is Adam. The learning rate is set to 0.001, determining the step size for adjusting the model’s weights. Sigmoid is used as the activation function for layer 1, while tanh is used for layer 2. These functions introduce non-linearity to the model’s computations. The dropout rate is set to 0.2, representing the proportion of neurons that dropped out during training to prevent overfitting. The model is trained for 300 epochs, which means the entire dataset is passed by the model 300 times during training. The batch size is set to 64, indicating the no. of samples employed in each training iteration before updating the model’s weights. The LSTM model consists of two layers, allowing it to capture complex patterns in the data. Each LSTM layer has 64 units or cells, enabling the model to learn and encode information. The recurrent dropout rate is set to 0.2, applying dropout specifically to the recurrent connections within the LSTM cells.

Table 3 Tuning hyper-parameters of the LSTM model.

The various hyperparameters adjusted during the training process of the LSTM model for parking space prediction.

Hyperparameter	Long short-term memory	
Learning rate	0.001	
Activation function	Sigmoid (layer 1), tanh (layer 2)	
Dropout rate	0.2	
Recurrent dropout	0.2	
Epochs	300	
Optimizer	Adam	
Batch size	64	
Number of layers	2	
Number of units	128	

Figure 8 depicts the loss that occurs among the training and validation data during the implementation process using the LSTM model for parking space prediction using the Melbourne dataset.

Figure 8 Training loses vs validation loses using the LSTM model in the Melbourne dataset.

The convergence of the model during the training process.

Figure 9 depicts the loss that occurs among the training and validation data during the implementation process using the LSTM model for parking space prediction using the Harvard dataset. As shown in Figs. 8 and 9, applying the LSTM model to the training dataset decreased the loss function. Moreover, it can be observed that the loss continued to decrease as the number of epochs increased, eventually reaching a minimum value. This indicates the success of the LSTM model in capturing patterns and improving the accuracy of the predictions for parking space occupancy.

Figure 9 Training loses vs validation loses using the LSTM model in the Harvard dataset.

The learning process of the model.

Figures 10 and 11 illustrate the comparison between the predicted values and the actual values for the nonlinear data points employing the LSTM model.

Figure 10 Results of the ARIMA and LSTM models (Melbourne dataset).

The figure compares the predictions generated by the ARIMA and LSTM models for the Melbourne parking dataset.

Figure 11 Results of ARIMA and LSTM models (Harvard dataset).

The figure compares the actual parking data with the predictions made by the ARIMA and LSTM models for the Harvard dataset.

Despite the reduction in losses, it has been proved that the model could not accurately capture the actual values. Notably, a critical disparity between the actual and predicted values indicates the model’s limitations in accurately forecasting parking space occupancy. In the next section, the predicted results of Phase 1 are fed into BPNN to optimize the error rate.

BPNN model

In Phase 2, the predicted result obtained from Phase 1 is fed into Phase 2 to reduce the errors and make accurate predictions. This phase integrates ARIMA and LSTM models and is optimized through BPNN. This hybrid model allows for the combination of forecasted values from both models employing a nonlinear activation function, effectively handling both linear and nonlinear associations among the outputs.

Table 4 represents the tuning of hyperparameters used for BPNN. The BPNN optimizes the output and minimizes errors by iteratively backpropagating errors throughout the neural network and readjusting the weights. The inputs given to the BPNN are scaled using the “MinMaxScaler()” function to enhance model performance. The training process involves 5,000 epochs, enabling the model to reduce errors and refine predictions progressively. Early stopping is implemented, and validation loss is monitored to prevent the model from learning noise in the training data and to overcome the problem of overfitting. A learning rate of 0.9 regulates the magnitude of weight adjustments during training. The next section will explore the final results of implementing the proposed hybrid model.

Table 4 Hyperparameters used for BPNN.

The key hyperparameters tuned during the training of the BPNN to optimize the proposed hybrid model.

Hyperparameters	Value	
Learning rate	0.9	
Activation function	Sigmoid (Layer 1), tanh (Layer 2), ReLU (Layer 3)	
Dropout rate	0.5	
Recurrent dropout	0.5	
Epochs	5,000	
Optimizer	Adam	
Batch size	32	
Number of layers	3	
Number of units	128	

Experimental results and discussion

This section discusses the performance metrics used to evaluate the model is discussed in detail. The results obtained during implementing the proposed hybrid model to predict available parking space are also discussed.

Performance metrics

The performance metrics used to evaluate the proposed hybrid model are mean squared error (MSE), mean absolute error (MAE), and root mean square error (RMSE). Regarding time series forecasting, accuracy is typically assessed by contrasting the predicted values to the actual values using various error metrics. In time series forecasting, we generally refer to the error metrics mentioned earlier to calculate the model’s performance and the closeness of its predictions to the actual values. The main focus is on the model’s error rate, which is evaluated as a key parameter to measure the performance of the hybrid model. MSE, MAE and RMSE are employed explicitly as comparison and evaluation parameters in this research (Dia, Ahvar & Lee, 2022). While other evaluation parameters are available, the chosen parameters provide meaningful insights into the model’s performance in error estimation.

Mean squared error: MSE measures the average squared difference among the predicted & actual values, providing a measure of the model’s overall error. It is calculated as Eq. (8) (Nhu et al., 2020).

(8) MSE=1n∑i=1n⁡(Yi−Y^i)2

where n represents the count of data items, Yi denotes the actual values, Y^i stands the predicted values, and Σ denotes the summation over all data points.

Mean absolute error: When you talk about MAE, you’re basically looking at how close the predicted values are to the actual ones. A lower MAE means that the model’s predictions are more accurate. It is calculated as Eq. (9).

(9) MAE=1n∑|Yi−Y^i|

where n represents the count of data items, Yi stands for actual values, Y^i stands for predicted values, and Σ denotes the summation of all data points.

Root mean squared error: When predicting parking space availability, the root mean square error (RMSE) helps you understand how close the predicted values are to the actual ones. A lower RMSE means the model fits the data well, leading to more precise predictions of parking space availability. So, a minimum RMSE value signifies better accuracy and reliability in forecasting parking space availability. It is calculated using Eq. (10).

(10) RSME=1n∑i=1n⁡(Yi−Y^i)22

where n represents the count of data items, Yi stands for actual values, Y^i stands for predicted values.

The performance metrics outlined in this section comprehensively evaluate the hybrid model’s predictive accuracy and efficiency. These metrics form the basis for comparing the proposed model with existing approaches in the next section.

Result and discussion

This section presents the results obtained from Phase 2 and discusses them in detail. This section also presented the statistical significance of the proposed hybrid model. Finally, the proposed hybrid model is compared with existing models.

Figure 12 represents the final output of Phase 2, which is used to predict the available parking space using the proposed hybrid model. The proposed hybrid model successfully minimizes the errors generated in Phase 1 and optimizes the results by making accurate predictions of the available parking space using different datasets. Both datasets have different environmental conditions and different load factors. The results proved that the proposed hybrid model reduces errors and predicts accurately. This result demonstrates the superior performance of the proposed hybrid model in accurately predicting parking space occupancy.

Figure 12 Actual and predicted result of the proposed hybrid model.

The figure compares the observed parking data with the predictions made by the proposed hybrid model.

Figure 13 represents the performance of the proposed hybrid model tested on two different datasets. It represents consistent scaling behavior for both datasets as the time taken increases prediction with the number of samples consistently maintained. As the data size increases, time slightly increases, but the proposed model can handle larger datasets efficiently.

Figure 13 Scalability performance test results on the Melbourne and Harvard datasets.

The performance of the proposed model in terms of scalability, tested on both the Melbourne and Harvard parking datasets.

Table 5 provides the results obtained during the implementation and validation of the proposed model in different training and testing distributions using the Melbourne dataset. Table 5 demonstrates the performance of the proposed hybrid model by minimizing the errors to improve parking prediction availability.

Table 5 Result based on different training and testing splitting ratio of the Melbourne dataset.

The performance of the proposed model using various training and testing data splits.

Training-testing percentage	Models name	MSE	MAE	RMSE	
70–30%	ARIMA	20.16	4.08	4.46	
70–30%	LSTM	25.53	4.30	5.10	
70–30%	ARIMA + LSTM	16.13	3.10	4.10	
70–30%	Proposed (ARIMA + LSTM + BPNN)	0.321	0.488	0.56	
80–20%	ARIMA	20.20	4.10	5.50	
80–20%	LSTM	25.53	4.10	5.16	
80–20%	ARIMA + LSTM	16.45	3.05	4.12	
80–20%	Proposed (ARIMA + LSTM + BPNN)	0.320	0.489	0.55	
90–10%	ARIMA	21.20	5.13	5.67	
90–10%	LSTM	27.67	5.80	5.90	
90–10%	ARIMA + LSTM	17.56	4.54	5.70	
90–10%	Proposed (ARIMA + LSTM + BPNN)	0.325	0.487	0.58	

Table 6 provides the results obtained during the implementation and validation of the proposed hybrid model in different training and testing distributions using the Harvard dataset. Tables 5 and 6 demonstrate that the proposed hybrid model performs optimally in different scenarios by using different datasets in different environments and minimizing the error rates. Table 7 represents the comparative analysis of different models used for parking space prediction.

Table 6 Result based on different training and testing splitting ratio of the Harvard dataset.

The performance outcomes of the proposed model using various training and testing data splits for the Harvard parking dataset.

Training-testing percentage	Models name	MSE	MAE	RMSE	
70–30%	ARIMA	24.20	6.80	6.32	
70–30%	LSTM	27.12	6.15	6.56	
70–30%	ARIMA + LSTM	17.21	3.15	4.50	
70–30%	Proposed (ARIMA + LSTM + BPNN)	0.311	0.478	0.56	
80–20%	ARIMA	20.20	4.10	5.50	
80–20%	LSTM	25.53	4.10	0.165	
80–20%	ARIMA + LSTM	17.45	3.20	4.10	
80–20%	Proposed (ARIMA + LSTM + BPNN)	0.310	0.412	0.54	
90–10%	ARIMA	24.20	5.50	6.18	
90–10%	LSTM	26.17	5.10	4.90	
90–10%	ARIMA + LSTM	17.16	4.54	5.10	
90–10%	Proposed (ARIMA + LSTM + BPNN)	0.315	0.447	0.52	

Table 7 Statistical significance of the proposed hybrid model.

The statistical analysis conducted to evaluate the significance of the proposed hybrid model.

Model comparison	MSE	MAE	RMSE	Statistical test	p-value	Significance (p < 0.05)	
Proposed hybrid model vs. ARIMA	0.321	0.488	0.56	ANOVA	0.02	Yes	
Proposed hybrid model vs. LSTM	0.321	0.488	0.56	ANOVA	0.04	Yes	
Proposed hybrid model vs. ARIMA + LSTM	0.321	0.488	0.56	ANOVA	0.01	Yes	
ARIMA vs. LSTM	0.321	0.488	0.56	ANOVA	0.15	No	

Table 7 highlights the performance metrics used for comparative analysis and the statistical analysis of the proposed hybrid model compared to the existing models, indicating improvements in the prediction accuracy. It presents a comparative analysis of different models used for parking space prediction, including the proposed hybrid model. Table 8 highlights the runtime of the different models, and the proposed hybrid model illustrates its efficiency for real-time applications in smart city development.

Table 8 Runtime of different models.

The computational efficiency of various models, including ARIMA, LSTM, and the proposed hybrid model, in terms of runtime.

Models	Training time (second/epoch)	Prediction time (μs/Prediction)	
ARIMA	1.18	0.02	
LSTM	1.19	0.02	
ARIMA + LSTM	1.85	0.03	
Proposed Hybrid Model	2.16	0.03	

Figure 14 represents the comparative analysis of the performance of the proposed hybrid model with different models. It represents the results of Tables 7 and 8 that help to understand the performance and runtime of the proposed hybrid model.

Figure 14 Comparative analysis of the proposed hybrid model with different models.

A comparison of the performance of the proposed hybrid model with several benchmark models, including ARIMA, LSTM, and others.

Table 9 represents the performance comparison of the proposed hybrid model with existing models for parking space prediction. The table compares the results of the proposed hybrid model using different datasets with the existing models.

Table 9 Performance comparison of the proposed hybrid model with existing models for parking space prediction.

A comparative analysis of the proposed hybrid model and several existing models, such as ARIMA, LSTM, and other relevant methods.

S. No.	Model	MSE	MAE	RMSE	
1.	LSTM + RF (Sadia et al., 2021)	–	3.10	4.64	
2.	LSTM (Xiangdong et al., 2019)	4.53	1.28	–	
3.	LSTM + RNN (Mudassar & Byun, 2018)	0.53	–	0.72	
4.	LR + ARIMA + NN (Zhao, Zhang & Zhang, 2020)	–	0.63	0.89	
5.	LSTM + ARIMA (Avşar, Anar & Polat, 2022)	–	0.89	1.13	
6.	CNN + LSTM (Xu et al., 2024)	–	13.301	21.156	
7.	GCU-GRU (Feng et al., 2023)	–	13.61	18.76	
G2RNN	–	7.35	10.56	
LSTM	–	7.96	11.35	
BPNN	–	8.17	11.63	
8.	Proposed hybrid model (ARIMA + LSTM + BPNN) Melbourne Dataset	0.32	0.48	0.56	
9.	Proposed hybrid model (ARIMA + LSTM + BPNN) Harvard dataset	0.31	0.47	0.56	

The proposed hybrid model (ARIMA + LSTM + BPNN) demonstrates superior performance with an MSE of 0.32, an MAE of 0.48, and an RMSE of 0.56 compared to the other models tested on the Melbourne dataset. It is again tested on the Harvard dataset and achieves performances with an MSE of 0.31, an MAE of 0.47, and an RMSE of 0.56. Figure 14 compares the proposed hybrid model’s performance with the existing model using performance metrics such as MSE, MAE, and RMSE. The proposed model shows the lowest error rate, indicating its improved performance. Figure 15 represents the performance comparison of the proposed model with existing models.

Figure 15 Performance comparison of the proposed model with existing models.

A side-by-side comparison of the proposed hybrid model against several existing models, such as ARIMA, LSTM, and other state-of-the-art techniques.

It is evident from Fig. 14 and Table 9 that the proposed hybrid model (ARIMA + LSTM + BPNN) outperformed by achieving a minimum error rate for predicting available parking space. By combining different forecasting techniques and leveraging the optimization capabilities of BPNN, the proposed model achieves more accurate predictions and outperforms the standalone models and other hybrid models. The proposed hybrid model is generalized and can be implemented on any parking dataset that compromises time series data, which contains linear and nonlinear data.

Conclusion

Parking space is an essential issue in society. Many researchers have proposed parking space prediction models. These models give efficient results whenever applied to small datasets, but the performance decreases if dataset size increases as the number of vehicles increases. Therefore, there is a need for a more scalable predictive model that can handle a large number of vehicles. This article proposed a novel hybrid model that combines ARIMA + LSTM + BPNN for available parking space prediction in smart cities. Phase 1 combines ARIMA and LSTM models optimized with BPNN at Phase 2 to improve prediction accuracy. The proposed model is scalable and can handle the situation if the number of vehicles increases. The model is implemented using the Melbourne time-series datasets.

Further, to validate and check the performance of the proposed hybrid model, it is implemented on another Harvard dataset. The proposed hybrid model achieves the minimum MSE, MAE, and RMSE values of 0.32, 0.48, and 0.56, respectively, on the Melbourne dataset. It achieves the minimum MSE, MAE, and RMSE values of 0.31, 0.47, and 0.56, respectively, on the Harvard Dataset. Its societal benefits include reducing traffic congestion and creating pollution-free and noise-free smart cities. The limitation of the proposed hybrid model is that it includes only a few features from both datasets. Future enhancements should explore incorporating external factors like weather and events and utilizing interpretation analysis methods like local interpretable model-agnostic explanations (LIME) and SHapley Additive exPlanations (SHAP) to enhance performance and provide detailed capabilities for the model’s predictions.

Supplemental Information

Supplemental Information 1 Data.

Supplemental Information 2 Code.

Additional Information and Declarations

Competing Interests

The authors declare that they have no competing interests.

Author Contributions

Anchal Dahiya conceived and designed the experiments, performed the experiments, analyzed the data, performed the computation work, prepared figures and/or tables, authored or reviewed drafts of the article, and approved the final draft.

Pooja Mittal conceived and designed the experiments, performed the experiments, analyzed the data, authored or reviewed drafts of the article, and approved the final draft.

Yogesh Kumar Sharma performed the experiments, analyzed the data, prepared figures and/or tables, authored or reviewed drafts of the article, and approved the final draft.

Umesh Kumar Lilhore conceived and designed the experiments, performed the experiments, analyzed the data, authored or reviewed drafts of the article, and approved the final draft.

Sarita Simaiya performed the experiments, analyzed the data, performed the computation work, prepared figures and/or tables, authored or reviewed drafts of the article, and approved the final draft.

Mohd Anul Haq analyzed the data, performed the computation work, prepared figures and/or tables, authored or reviewed drafts of the article, and approved the final draft.

Mohammed A. Aleisa performed the computation work, prepared figures and/or tables, authored or reviewed drafts of the article, and approved the final draft.

Abdullah Alenizi performed the computation work, prepared figures and/or tables, authored or reviewed drafts of the article, and approved the final draft.

Data Availability

The following information was supplied regarding data availability:

The raw data and code are available in the Supplemental Files.

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
