# Peer review of "Hybrid parking space prediction model: integrating ARIMA, Long short-term memory (LSTM), and backpropagation neural network (BPNN) for smart city development"

_PeerJ Computer Science, doi:10.7717/peerj-cs.2645_

## Round 0.1 · original submission · Major Revisions

Please address the comments of all four reviewers. Given the fact that several of the Reviewers are quite critical, there is no guarantee your article will be Accepted. Please address the question of lack of novelty

Reviewer 1 ·

Basic reporting

English and structure is fine.
Sufficient literature is not discussed. Problems, available solutions and research gap is not discussed.
Introduction is not written well. The authors should discuss about the problems, impact and possible solutions.
I could not find a discussion related to smart city development and parking prediction in the article.
Critical writing is missing in the paper.

Experimental design

Aim and contributions are not clear.
Research gap is not discussed.
Contribution should be re-evaluated and modified. The current list of contributions are not correct.
In page 11, there are question marks (?) in equations values or parameters.
Implementation of hybrid model should be elaborated. The current discussion of stages is very vague.

Validity of the findings

Dataset is not discussed sufficiently.
5000 epochs would lead to overfitting. But there is no discussion related to this aspect.
Discussion should evaluate the results critically which is also missing.

Reviewer 2 ·

Basic reporting

This paper proposes hybrid a smart parking space prediction model that combines ARIMA and LSTM models. The proposed model is hybridized using a BPNN. Basically, the work is interesting. However, there are some concerns should be addressed.
1. The prediction model is not presented clearly at all. There is no justification of the use of ARIMA, LSTM and BNPP. How many historical observations are fed into the model? How long into the future does this model predict the number of vacant parking spaces? What are the attractive properties of these models for processing vacant parking space time series data?
2. I cannot see the proposed model in section Materials & Methods. This section is only the background introduction, where ARIMA, LSTM and BNPP are introduced separately. In addition, this section seems to be too lengthy. It would be better to shorten the text, possibly into one half, so that readers can go straight into the essence of this paper.
3. Also, the authors are suggested to present the proposed model in section Materials & Methods.
4. There is no discussion on the cost of the proposed method. What is the runtime? Please include such discussions. The authors are suggested to compare the runtimes of different methods.
5. Are the reported precision differences statistically significant?
6. The authors should test the model with at least one more dataset collect from another parking lot to verify the generalization of the model.
7. To have an unbiased view in the paper, there should be some discussions on the limitations of the proposed method.
8. The methods used for comparison are too old. The authors are suggested to add more state-of-the-art comparison algorithms are recommended.
9. Some recent closely related references on vacant parking space availability prediction are missed., e.g. 10.1109/TVT.2023.3266224, 10.1007/s40747-022-00700-1, 10.1109/MITS.2020.3014131, 10.1049/iet-its.2018.5031.

10. The manuscript needs proofreading and further revise. There are some typos and grammar errors.

Experimental design

See “Basic reporting”

Validity of the findings

See “Basic reporting”

Additional comments

no comment

·

Basic reporting

The article is well-structured, providing a clear and concise introduction to the importance of parking space prediction in smart city development. The literature review is thorough, covering relevant studies and highlighting the gap addressed by this research. Figures and tables are appropriately used to illustrate the model's performance and the data sources.

Experimental design

The experimental design is solid, combining ARIMA, LSTM, and BPNN to create a hybrid model for parking space prediction. The choice of these models is well-justified, and the methodology is described in detail, enabling reproducibility. However, additional information on the data preprocessing steps and hyperparameter tuning would enhance the transparency of the experimental setup.

Validity of the findings

The findings are valid, supported by comprehensive statistical analysis and comparisons with baseline models. The hybrid model demonstrates superior performance in predicting parking space availability, making a significant contribution to smart city initiatives. The discussion of results is insightful, although a deeper analysis of the potential limitations and the model's scalability to different urban environments would be beneficial.

Reviewer 4 ·

Basic reporting

1. The language is clear.
2. The literature review should discuss existing studies closely related to this work. In addition, it should indicate the existing research gaps that motivate this study. Please focus on literature related to parking demand prediction.
3. Article structure is good. Please double-check the citations of tables and figures in the article. For example, "Figure 3" in Line 313 should be "Figure 2".
4. Results are presented well.
5. Equations 2 - 8 are not readable.
6. Algorithms should be presented using pseudocode.

Experimental design

1. Topic fits this journal.
2. Research question is well defined. However, the introduction should introduce the meaning of this study. Too much unrelated content in the introduction. The statements of how this research fills knowledge gap are lacking.
3. Investigation is conducted rigorously.
4. Most of the methods are presented in detail.

Validity of the findings

1. The methodology lacks novelty. The method used to combine ARIMA and LSTM is shallow.
2. LSTM is capable of capturing both linear and non-linear time series patterns. Please explain the necessity of ARIMA in this model.
3. LSTM networks do employ a variant of backpropagation called Backpropagation Through Time (BPTT). Please explain why the BPNN is needed.
4. The hyperparameter tuning process is lacking. A fine-tuned LSTM could perform better.
5. Should include more deep learning-based benchmark models such as CNN, GCNN, Transformer, etc.

---

## Round 0.2 · Major Revisions

As you can see, 3 out of 4 of the reviewers still have significant requests which must be addressed or the article will be unable to progress.

Reviewer 1 ·

Basic reporting

In Introduction, it would be good to discuss some relevant methods and their problems to understand the need for novel model

In contribution, first point is valid, but not the other two.

Experimental design

Critical analysis is still missing when analysing other literature or results.

Relevant literatures mentioned in Table 1 can be discussed in detail. Discuss problem with these methods in terms of scalability and accuracy.

Validity of the findings

Results are fine, but analysis and discussion should be improved.

Discussion on methods and the reason for performance improvement should be discussed.

Reviewer 2 ·

Basic reporting

I found the section on related works to be disorganized. It would be beneficial to cite the most recent papers and categorize them based on different methods employed. New cited works should also be summarized in Table 1.
The quality of the figures is low, the resolution is too low. Please do not go below 300 dpi.

Experimental design

The author did not fully consider my eighth comment. I still suggest that the author should compare the proposed model with other state-of-the-art models to verify the superiority. Now, many models come with their source code provided, making it not very complex to reproduce these models.

Validity of the findings

no comments

·

Basic reporting

No Comment

Experimental design

No Comment

Validity of the findings

No Comment

Reviewer 4 ·

Basic reporting

1. Language, article structure, and result presentation are good.
2. Literature review needs to be improved. The first two paragraphs of the Literature Survey section have some duplicate statements such as Line 122 "Several researchers contributed in this area to solve parking-related problems..." and Line 126 "Several researchers work in this area".
3. Table 1 looks good. I suggest using one or two paragraphs to summarize and discuss the content in Table 1.
4. In Figure 8, why are some observed values (observed parking occupancy) below 0?

Experimental design

5. Topic fits this journal. Research question is well-defined and relevant.
6. Line 502. 300 epochs are too many according to Figure 15 and 16.

Validity of the findings

7. Please compare the performance (MSE, MAE, RMSE) of the proposed model and additional deep learning-based benchmark models (such as CNN, GCNN, and Transformer) using the Melbourne Dataset and Harvard Dataset.

---

## Round 0.3 · accepted · Accept

Dear Authors,
Your paper has been accepted for publication in PEERJ Computer Science. I ask that you make minor changes to your manuscript based on the reviewers' comments, before uploading final files. Thank you for your fine contribution.

Reviewer 2 ·

Basic reporting

No more comments

Experimental design

No more comments

Validity of the findings

No more comments

Additional comments

The authors have addressed all my previous concerns. I suggest accepting the paper.